# Nutrition Transition and Cancer

**DOI:** 10.3390/nu12030795

**Published:** 2020-03-18

**Authors:** Franco Contaldo, Lidia Santarpia, Iolanda Cioffi, Fabrizio Pasanisi

**Affiliations:** Clinical Nutrition and Internal Medicine, Department of Clinical Medicine and Surgery, Federico II University Hospital of Naples Via Pansini, 580131 Naples, Italy; contaldo@unina.it (F.C.); lolanda.cioffi@unina.it (I.C.); pasanisi@unina.it (F.P.)

**Keywords:** nutrition, non-communicable diseases, cancer, human health

## Abstract

Urbanization, population aging, and climatic changes have mostly contributed to nutrition transition and, consequently, to effects of food habits on the epidemic of chronic non-communicable diseases (NCDs), especially cancer. Climatic changes are negatively affecting crop production, particularly biodiversity, leading to reduced food choices and, consequently, nutritional value and the protection conferred from consumption of a variety of nutrients essential in a healthy diet. This brief review analyzes the possible link between rapid demographic changes, climatic and environmental crises, and the current food system as possible factors contributing to the role of nutrition transition in the onset of cancer.

## 1. Introduction

The relation between cancer (at least some types) and nutrition has been widely discussed in the literature [1,2,3]. Foodstuffs, food preparation and cooking, macro- and micronutrient composition, contaminants, additives, etc., along with the entire food system have been evaluated to identify the role of food in increasing or decreasing the risk for developing different types of cancer [4,5].

In the past 50 years, patterns of human food habits and food production have dramatically changed, both in developed and developing countries. From local fresh foods—vegetables, tubers, and animal-sourced foods—and home-cooked basic commodities, we have moved to packaged and processed ready-to-eat or ready-to-heat foods [6,7]. These quick modifications, forming part of the so-called “nutrition transition”, can severely affect the adaptive biological mechanisms of the human species [8,9] and the environment [10]. Accordingly, recent studies in the literature have described the devastating effects of climate change on human health [11,12,13]. The first and probably more deleterious negative effect of climate change concerns the biodiversity of both animal and plant species. Low biodiversity can limit food choices and consequently affect the intake of macronutrients (carbohydrates, protein, lipids), micronutrients (vitamins and minerals), antioxidants, and other protective substances. For instance, this is the case for crop production, which is negatively influenced by climate change and, in turn, affects the human diet and health status.

So far, medical progress has changed the natural history of several diseases, in particular, of non-communicable diseases (NCDs) such as obesity, cardiovascular diseases, and cancer, whose clinical patterns and evolution have shifted from being acute to chronic, thus requiring long-term interventions in lifestyle and radical social transformations. In the meantime, demography, a well-studied sociobiological variable, has rapidly changed. In fact, the number of humans living on the planet has significantly increased by about 4 times (from about 1.5 to more than 7.5 billion) during the last century while life expectancy has doubled within that same period (from 40 to about 80 years). Overall, these dramatic changes in the characteristics of our species have prompted scientists to identify a new era: the Anthropocene, a geological epoch characterized by the prevalence of a single species, *Homo sapiens*, with consequent environmental anthropization.

Over the years, human activities have had a substantial disruptive effect on Earth’s systems [11,12,13]. Indeed, as thoroughly described by Willet et al. [14], the most prominent health issue in the Anthropocene era is to simultaneously face the syndemic—or synergy of epidemics—of chronic NCDs and climate disasters [10]. This is not the first time that natural disasters have been linked to human health, but the actuality of this link should consist of a strict ubiquitous relation between the huge anthropic presence and natural disasters. To date, the main accepted laws for explaining human adaptation in the Paleolithic and Neolithic ages have been genetic drift and adaptive developmental plasticity. Both laws have been recently investigated and evaluated from an epigenetic view [8,9]. These epigenetic studies have reported that the expression of some genes are modulated early to allow better and immediate adaptation to the outside living environment with better outcomes for survival. Currently, increased life expectancy, associated with more frequent and rapid migrations than those performed during previous ages, has revealed a potential late health hazard for this immediate protective function, like in the case of obesity and related diseases such as breast cancer.

In this short review, we would like to address the link between food, health, and lifestyle from an ecological point of view. Therefore, climatic changes, demography, the food system, and epigenetic influences are reviewed herein in order to suggest a comprehensive ecological approach to the relation between nutrition and cancer disease.

## 2. Climate Changes and Decline in Biodiversity

Today, biodiversity decline represents perhaps the most alarming boundary effect linked to climate change. Currently, the decline in food choices due to poor biodiversity is attributed to the gradual and complex passage from hunting to foraging (agriculture and livestock). This gradual transition began 10,000–12,000 years ago during the transition from the Paleolithic to the Neolithic age [15]. It has been estimated that of the 3000 seed-bearing plants available in the Paleolithic, 200 were grown in the early Neolithic. To date, however, only 13 are preferentially produced, with 4 types being particularly prevalent: maize, wheat, rice, and sugar cane. Similarly, from about 50 species of animals raised in the early Neolithic, only chicken, pig, and cattle are currently consumed in westernized societies. Additionally, food processing (and ultra-processing), which is the final stage of the current food system, has negatively contributed to changes in food habits, linking persistent and unnatural food consumption to the development of chronic degenerative diseases [16,17,18]. This socioeconomic change has been called “the retail revolution”, characterized by the disappearance of fresh markets and the development of numerous, but small, food retailers [19]. This tremendous and unrecognized reduction in food choice, as part of the environmental impact, may strongly reduce micronutrient consumption, resulting in a dramatic increase of selective malnutrition as well as in poor intake of protective antioxidant natural agents. The consequences of these chronic nutritional deficiencies, associated with chronic exposure to a variety of pollutants, are easily predictable. Although the Neolithic revolution, characterized by a shift from foraging to farming, let people have a more regular, higher-energy food supply, it caused changes in the food system as well as in dietary diversity, food preferences, and social behaviors. From the Neolithic age, as a matter of fact, social influences have affected human health, nowadays having uncontrolled effects.

The demographic boom reached at the end of the last century (from less than 1.5 billion humans at the beginning of the last century to more than 7.5 billion at present) rapidly changed the characteristics of the planet, introducing humans in the Anthropocene era [14]. Briefly, the Anthropocene is characterized by the prevalence of *Homo sapiens* over all other species, resulting in a serious disequilibrium in the environment. One of the main and potentially most negative determinants of the Anthropocene is the uncontrolled human food chain. In both developed and developing world countries, the obesity epidemic represents an anthropological symbol of the fast and negative modifications in food habits (nutrition transition). Increases in crop yields and improvements in production practices have contributed to reductions in hunger and improved life expectancy [1,2]. However, these health benefits are being offset by global shifts toward unhealthy diets that are high in calories as well as heavily processed and animal-sourced foods, which enhance the risk of developing obesity. Transitions to unhealthy diets are not only increasing the burden of obesity and diet-related NCDs but are also leading to environmental degradation. Therefore, the impact on the environment as well as on food health, in terms of both prevalence and incidence of chronic NCDs, has been unavoidable and expected.

Recently, the *Lancet* used the term “syndemic” for the second time (with the first being in the context of the HIV epidemic) to associate climate change (a disease of nature) with the obesity epidemic (a human disease), drawing a definite linking between the health (future) of the planet to the health (future) of our species [14]. Therefore, encouraging people to eat more efficiently with regard to the food chain, by consuming less (but not necessarily excluding) meat and more plant-based foods, may be helpful for increasing sustainability and reducing the environmental costs of the food (production) system.

## 3. Nutrition Ecology

In 2003, a supplement was published by the *American Journal of Clinical Nutrition* that is fully devoted to promoting the nutrition ecology discipline as an interdisciplinary science with special consideration of the effects of nutrition on health, environment, society, and economy [20]. Food production is one of the largest contributors to both climate and environmental changes. Diet can link human health and environmental sustainability. Clearly, a healthy diet is largely more sustainable than an unhealthy one because it mostly consists of plant-based foods such as fruit, legumes, unsalted nuts, whole grains, unsaturated oils, etc., with low amounts of red meat, fish, poultry, cheese, and refined grains [16].

Nutrition ecology is part of a comprehensive project, also recently sponsored by the *Lancet* Commission [6,14], aimed to protect the environment and human health by strongly supporting sustainable development and a healthier urbanization process, such as healthy urban design, transport, and the so-called “compact city”. Although this urbanization is not the topic of this review, this approach facilitates spontaneous physical exercise and healthy overall individual wellbeing, results in reduced particulate emissions due to reduced motor vehicle emissions [19,20].

## 4. Food System

The alimentary chain, due to its complexity, is called the “food system”. One of the more comprehensive definitions of the “food system” was reported by Popkin et al., as follows: “the broader system defined by the activities, infrastructures and people involved in feeding the global population (e.g.,: the growing, processing, distribution, consumption and disposal of foods). It includes the network of processes by which institutions, organizations, and individuals transform inputs into foods and natural ingredients into the food that we eat (including seeds, fertilizers, chemicals, pharmaceuticals, …)” [12].

The negative impact of the present food system on the ecosystem was dramatically reported in a 2015 study commissioned by the Rockefeller Foundation and the *Lancet*: [8] “Food production commands up to 25% of net primary productivity on land affecting climate stability. Nowadays, four out of nine climate boundaries (climate change, deforestation, biodiversity loss, nitrogen and phosphorus abuse in fertilization) have been directly overcome. Another five (ozone depletion, desertification, wetland loss, urbanization, coastal reef damage) are at high risk”. As a result, the overconsumption of unhealthy food occurs at the expense of the resilience of the planet. Still, in 2014, the Rockefeller Foundation and *Lancet* Commission reported that out of the entire human population, 600 million (8.2%) were obese, 1.7 billion (17.8%) were overweight, 800 million (11.0%) suffered from hunger, and 2 billion (27.4%) had nutrient deficiencies. In conclusion, partly due to the devastating effects of the actual food system on the ecosystem, about 60% of the whole human population suffer from malnutrition (excess, deficit, mixed, selected deficiencies) and associated diseases, including some types of cancers [8,21].

## 5. The killer: Which Nutrient(s)?

The complexity of the present food system, involving food macro-/micronutrient content, method of Production, transformation, packaging, etc., cannot allow us to identify a “prevalent or unique killer” responsible for harmful effects on health. In the late 1960s, there was scientific debate concerning the unhealthy effect of a dietary excess of saturated fats (sponsored by Keys, who suggested adopting the Mediterranean diet and lifestyle) or of simple sugars (proposed as a killer by Yudkin, who suggested a low-carbohydrate (CHO) diet). However, at that time, the food system was largely different from that now [17].

Currently, the food consumed in many parts of the world is shifting from purchases made at local fresh markets to packaged and processed or ultra-processed ready-to-heat or ready-to-eat foods corresponding to more than 80% of calorie intake, at least in the United States [15,18]. Furthermore, Myers et al. pointed out that future food production will be affected by climate change with secondary adverse consequences to health. The authors hypothesized that by 2050, climate-related deaths will be mostly due to reduced fruit and vegetable production [21]. Therefore, poor availability of fresh food, like fruit and vegetables, associated with high consumption of ultra-processed foods may represent a negative scenario for the prevention of all NCDs, including cancer.

Ultra-processed foods contain multiple refined ingredients and are manufactured to be hyperpalatable, affordable, and with a long shelf life. [15,22]. In addition to the poor nutritional value (high in calories, saturated fat, refined CHO, and salt, low in fiber), there are also concerns regarding the industrial processes involved, due to the use of contaminants and food additives [23,24]. Recently, Schnabel et al. summarized different plausible risk factors linking the overconsumption of ultra-processed foods with cancer, as follows: (1) the high temperature of food processing may lead to formation of contaminants; (2) acrylamide may have a carcinogenic and genotoxic effect; (3) the overconsumption of processed meat could predispose people to a high risk of stomach and colorectal cancer; (4) additives like titanium dioxide may be associated with increased risk of intestinal inflammation and carcinogenesis; (5) emulsifiers and abuse of artificial sweeteners may alter the gut microbiota, causing predisposition to colorectal cancer; and (6) food packaging may facilitate contact with toxic chemicals like bisphenol A, which is supposed to have endocrine-disrupting properties [23]. In conclusion, the overconsumption of ultra-processed foods may increase neoplastic risk due to exposure to potentially carcinogenic chemicals. Furthermore, this might negatively modify individual food habits by potentially worsening the unhealthy Western (transition) diet. Therefore, it could be useful, from a clinical point of view, to identify nutritional markers to assess the adequacy of an individual’s diet. This task appears quite difficult considering the multi-ethnicity of globalized communities, the variety of utilized foods and their methods of preparation, and the widespread use of processed foods. Generally, the nutritional adequacy of an individual’s diet can be assessed by using the dietary reference values (DRVs) [25,26,27].

## 6. Dietary Reference Values (DRVs)

The dietary reference values (DRVs) can be used to identify nutrient intakes that are relevant for planning dietary treatments and for research purposes in individuals as well as in population groups [27].

They include different terms such as the population reference intake (PRI), average requirement (AR), adequate intake (AI), and reference intake (RI) ranges for macronutrients [27]. The PRI is the level of nutrient intake that is adequate for the majority of people in a population group, whereas the AR is the intake level that is adequate to meet the physiological needs of half of the individuals in a population. When there is insufficient scientific evidence to estimate the AR or PRI, the AI is established by estimating the intake of an apparently healthy population group that is assumed to have adequate intake. Finally, the RI is the intake range for macronutrients, expressed as a percentage of the energy intake, and corresponds to the range that is adequate for maintaining health.

The RI introduces tremendous variability in designing personalized diets (according to the multi-ethnicity of most societies) [25]. Based on the RI ranges, a healthy diet should provide 45–65% of energy intake from CHO, 20–35% from fat, and 10–35% from protein. In general, the elimination or substantial restriction (or abuse) of a food or food group could affect individual health unless compensatory choices are made. In this context, some “nutrients of concern” have been identified for which monitoring can help in order to assess the nutritional adequacy of the diet and make targeted corrections. Nutrients of concern are nutrients that are considered to be typically overconsumed or underconsumed based on the DRVs, which is responsible for detrimental effects on health. Generally, “nutrients of concern” that need to be monitored are vitamin D, vitamin B12, vitamin A, vitamin B1, and folate, given their relationship with calcium, potassium, and iron [26,27]. For example, vitamin D promotes the absorption of calcium and is important for bone growth and maintaining healthy bones [28]. Calcium is a key mineral involved in bone health as well as in vascular, muscle, and nerve functions, besides its role in intracellular signaling and hormone secretion [29]. Getting enough potassium can positively and significantly cut the risk of high blood pressure, heart disease, and stroke [30,31]. Finally, iron is essential for carrying oxygen to all tissues in the human body [27].

## 7. Latency Period between Exposure to Transition Diet and Cancer Disease Diagnosis

Exposure to dietetic carcinogen agents is a chronic process. Hence, dietetic interventions require planned and preventive long-term actions based on the formation, prioritization, and implementation of national public health guidelines. Nevertheless, some authors suggest a plausible latency period between dietary intake, based on food habits, and the diagnosis of cancer. Shield [32] estimates an average latency period of 10 years between insufficient fiber intake, red and processed meat consumption, and insufficient fruit and vegetable consumption and the development of cancer. Similarly, Grundy et al. [33] observed that the latency period ranged from 6 to 20 years for cancer development due to inadequate fiber intake, from 10 to 14 years for colorectal cancer due to red/processed meat consumption, and from 4 to 9 years for oncological disease related to insufficient fruit and vegetable intake. According to a recent systematic review and meta-analysis, low intake of fruits, vegetables, and whole grains has been described as the largest contributor to cancer risk attributable to diet [34]. However, the risk may also be determined through indirect mechanisms such as obesity. Colorectal and lung cancers are the most frequently diagnosed neoplasia attributable to diet [35].

## 8. Transgenerational Effects of Dietetic Factors

Increasing evidence now shows that phenotypic variation can sometimes occur when environmental factors and genetic variants in previous generations create an epigenetic state that persists across generations in individuals who are not directly exposed to that environment or who do not inherit the original genetic variant [18]. Diet may, therefore, have transgenerational effects through epigenetic reprogramming of the gamete-carried determinants and, through this pathway, facilitate the onset of cancer disease.

In fact, cancer is a genetic disease characterized by inherited sporadic mutations in genes that maintain tissue homeostasis, control the cell cycle, or regulate apoptosis. Cancer is also an epigenetic disease with mutations in chromatin remodeling enzymes and epigenome or microRNA (mi-RNA) alterations, both possibly due also to nutritional factors. There is also evidence, as already mentioned, of the transgenerational effects of poor nutrition in diets: potential molecular mechanisms include methylation of gametes in the paternal and/or maternal lineage [18]. Indeed, unfavorable environmental conditions are able to influence genetic programming through demethylation of specific regions of the genome and new methylations of other sites. Despite being established by the same methyltransferase complex, the sperm methylome operates very differently from the oocyte methylome. In oocytes, the hyper- and hypomethylation of domains is respectively correlated with active transcription units and intergenic or inactive genomic regions, while in the male germline, methylation appears to be established indiscriminately. However, at present, the underlying transgenerational epigenetic mechanisms remain largely unknown [36].

A series of plant- and animal-based foods containing nutrients with protective anticancer action is summarized in the comprehensive work of [18].

Nutrients and bioactive food components can act either directly by inhibiting epigenetic enzymes that catalyze DNA methylation or histone modifications, or indirectly by altering the availability of substrates necessary for those enzymatic reactions. This, in turn, modifies the expression of critical genes associated with physiological and pathological processes, including embryonic development, aging, and carcinogenesis [36,37]. There is growing evidence that folate modulates anticarcinogenic properties through epigenetic changes, as low folate intake has been associated with DNA hypomethylation and increased risk of colorectal and pancreatic cancers [37,38]. Diets rich in fruits and vegetables and containing natural antioxidants may protect against cancer.

Certain dietary components (such as curcumin and retinoic acid) [39] may protect against cancer through miRNA regulation.

The potential epigenetic effects of several other nutritional components, mostly derived from vegetables (polyphenols, isothiocyanates, diallyl disulfide), consist of reduction in the DNA hypermethylation of critical genes, finally resulting in tumor suppression [40]. As a matter of fact, both animal- and plant-based foods may have protective roles provided by different food groups present in the diet. Dietary patterns, in addition to specific nutrients, also influence behavior and phenotype in offspring [41].

The overall conclusion is in favor of a semi-vegetarian diet which also considers a reasonable proportion of animal-sourced food, quite close to the traditional Mediterranean diet style. The Mediterranean diet, rich in vegetables and fresh fruits, whole grains, and seafood, is associated with a reduced incidence of cancer: the risk of breast cancer has been reported to be reduced in women who eat a Mediterranean diet supplemented with extra-virgin olive oil and mixed nuts [41].

Therefore, greater attention should be paid to the overconsumption of processed and ultra-processed meat, as well as of salty food in general, and to packaging procedures to reduce the risk of cancers.

## 9. Adiposity and Cancer

An association between excess body fat and cancer has been observed in all populations [42,43]. According to a recent report by the International Agency for Research on Cancer (IARC), there is sufficient evidence linking excess body fatness in adulthood to at least 13 different types of cancer, including esophageal adenocarcinoma, breast cancer (after menopause), colon and rectal, endometrial (corpus uterus), gallbladder, gastric, kidney (renal cells), liver, ovary, pancreas, thyroid, meningiomal, and multiple myeloma [42].

Generally, obesity is associated with substantial metabolic and endocrine abnormalities, including alterations in sex hormone metabolism, insulin and insulin-like growth factor (IGF) signaling, and adipokines or inflammatory pathways. Specifically, there is strong evidence for the role of chronic inflammation in mediating the obesity–cancer relationship, while the role of insulin and IGF appears to be moderate [44,45]. Hence, the increased worldwide prevalence of obesity in children and adolescents is expected to potentially increase the risks of developing various types of cancer later in life, due to excess body fat [46]. Since excess body fat and cancer share notable contributing factors, including poor diet and physical inactivity [47], the prevention of overweight and obesity in children, adolescents, and young adults should be highly emphasized to contain the obesity epidemic and any further increases in the burden of cancer linked to excess body fat.

## 10. Conclusion: Nature Versus Nurture

The relationship between cancer and climate change seems, without any doubt, to be a consequence of the current unsustainable food system, developed in recent decades. Cancer may even be considered a consequence of the transformation of nature by nurture, i.e., by how nature has been irreversibly modified by pervasive human activities. An ecological approach, including new food production and sustainable technologies in addition to medical treatments, appears urgent and unavoidable. Nutrition ecology should represent the comprehensive framework within which any dietetic advice should be considered, allowing the responsible consumption of any food group derived from either vegetable or animal sources while also considering individual culture, preferences, and ethnicity.

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
