# Peer review of "Nutrition Transition and Cancer"

_nutrients, 2020, doi:10.3390/nu12030795_

Round 1

Reviewer 1 Report

In this interesting review article by Contaldo et al. the authors touch upon the repercussions of the observed shifts in ecology and nutrition on human health. It provides a good historical perspective of how diets have changed, and some explanation of the factors that have fueled these changes. But, the evidence provided to support the authors arguments are tenuous and would need to be better supported by a more thorough literature review before publication.  

Major Points:

Extensive English revision is necessary to render this review article more readable. Some representative examples of the problems are highlighted below in the minor points. Yet, these are only examples. The text is littered with poorly phrased, or sometimes, wrongly phrased sentences. This detracts from the very interesting message the authors are trying to communicate. CHO needs to be defined. It is discussed in many different contexts without ever having had a definition of what it is provided. The arguments need to be better supported. In some cases, minimal additional information is necessary even to understand the point that the authors would like to get across. For example, in Section 6 regarding AMDR. There needs to be a more ample discussion of what nutrients of concern are, why these particular nutrients, what information do they provide? Again, in section 7, which cancers are being referred to in the Shield et al report. It is not sufficient just to provide the references. The point of reading a review is to gain an overall view of a specific topic, which, should be supported by references, but clear and accessible in the review. This review is not meeting this criteria in all sections. In section 8 what are the trans-generational determinates? Methylation of gametes? Methylation does exist in gamete, are specific alleles differentially methylated? And in response to what? Also, the mechanism by which dietary fonts of metabolites involved in epigenetic is unclear to me. Why are they protective, does this protection extend to all cancers, or a particular subset. Are these specific to trans-generational inheritance? That is what the title of the section implies. But, the cited article refers to protective effects of nutrition on epigenetics beyond the gamete. I find that replication of the table from an already published review in the same family (MDPI) is inappropriate. The table should include new and additional data or references in order to add something to the field. Otherwise just cite the original reference.

Minor Points:

Spurious use of capital letters such as: nutrition transition in line 10, cancer and nutrition on line 20, globalization on line 41, syndemic on line 45, and food system on lines 52-53. “This review shortly analyses…” (line 14) should read This brief review analyses… Wrong selection of adjectives that render the phrase grammatically incorrect such as: “The number of humans living on the planet has incredibly increased,….” A better word selection would be “has significantly increased,…” References are missing, such as the one in support of the sentence beginning on line 127 and finishing on line 129. Abbreviations such as those for the minerals indicated on line 177 should be explained, or, preceded by the full word, Calcium (Ca) for example.

Author Response

Reviewer 1

Comments and Suggestions for Authors

In this interesting review article by Contaldo et al. the authors touch upon the repercussions of the observed shifts in ecology and nutrition on human health. It provides a good historical perspective of how diets have changed, and some explanation of the factors that have fuelled these changes. But, the evidence provided to support the authors arguments are tenuous and would need to be better supported by a more thorough literature review before publication.  

Major Points:

Extensive English revision is necessary to render this review article more readable. Some representative examples of the problems are highlighted below in the minor points. Yet, these are only examples. The text is littered with poorly phrased, or sometimes, wrongly phrased sentences. This detracts from the very interesting message the authors are trying to communicate.

The authors thank the reviewer for  interesting comments and useful suggestions.

As far as the English language, the manuscript has been  revised by MDPI, as attested by the attached certificate. Anyway further revision has been obtained.

CHO needs to be defined. It is discussed in many different contexts without ever having had a definition of what it is provided.

The acronym CHO stands for carbohydrates, now defined in the text (line 138, page 3)

The arguments need to be better supported. In some cases, minimal additional information is necessary even to understand the point that the authors would like to get across. For example, in Section 6 regarding AMDR. There needs to be a more ample discussion of what nutrients of concern are, why these particular nutrients, what information do they provide?

Nutrients of concern are nutrients that are typically either over-consumed or under-consumed based on the AMDR, being responsible for detrimental effects on health. Identified “nutrients of concern” to be monitored are vitamins : D, B12, A, B1 and folate ; minerals : calcium, potassium and iron. (new Ref 32). For example, vitamin D promotes the absorption of calcium and is important for bone growth and maintaining healthy bones.(new Ref 33). Calcium is a key mineral involved in bone health as well as in vascular, muscle and nerve function, besides intracellular signalling and hormone secretion (new Ref 33,34). Getting enough potassium, usually marker of an adequate amount of vegetable foods intake,  can positively reduce the risk of high blood pressure, heart disease, and stroke (new Ref 35,36). Iron is essential for carrying oxygen to all tissues of human body (new Ref 32). Added in the text (line 175 – 182, page 4)

32.Health.gov, Dietary Guidelines 2015-2020, https://health.gov/dietaryguidelines/2015/ guidelines/, accessed 11/27/17 US Food & Drug Administration, “Changes to the Nutrition Facts Label,” https://www. fda.gov/ Food/GuidanceRegulation/GuidanceDocumentsRegulatoryInformation/LabelingNutrition/ucm385663.htm

33.Office of the Surgeon General (US). Bone Health and Osteoporosis: A Report of the Surgeon General. Rockville (MD): Office of the Surgeon General (US); 2004.

34.Beto, Judith A. “The Role of Calcium in Human Aging.” Clinical Nutrition Research 4.1 (2015): 1–8.

  1. The American Journal of Clinical Nutrition, “Sodium and potassium intakes among US adults: NHANES 2003–2008,” http://ajcn.nutrition.org/content/96/3/647.long,

36.Weaver, Connie M. “Potassium and Health.” Advances in Nutrition 4.3 (2013): 368S–377S.

Again, in section 7, which cancers are being referred to in the Shield et al report. It is not sufficient just to provide the references. The point of reading a review is to gain an overall view of a specific topic, which, should be supported by references, but clear and accessible in the review. This review is not meeting this criteria in all sections.

Low intake of fruit and dietary fibre has been described as the largest contributor to cancer risk attributable to diet .(new Ref 39). The risk may also be determined through indirect mechanisms such as obesity. Colorectal and lung cancers are the most frequently diagnosed neoplasia  attributable to diet .(new Ref 40). Added in the text (line 192 - 195, page 5)

39. Schwingshackl L, Schwedhelm C, Galbete C, Hoffmann G. Adherence to Mediterranean Diet and Risk of Cancer: An Updated Systematic Review and Meta-Analysis. Nutrients. 2017 Sep 26;9(10). 40. Kim M, Park K. Dietary Fat Intake and Risk of Colorectal Cancer: A Systematic Review and Meta-Analysis of Prospective Studies. Nutrients. 2018 Dec 12;10(12).

In section 8 what are the trans-generational determinates? Methylation of gametes? Methylation does exist in gamete, are specific alleles differentially methylated? And in response to what?

Increasing evidence now shows that phenotypic variation can sometimes occur when environmental factors and genetic variants in previous generations create an epigenetic state that persists across generations in individuals who are not directly exposed to that environment or who do not inherit the original genetic variant  (new Ref 14). Indeed unfavourable environmental conditions are able to influence genetic programming through demethylation of specific regions of the genome and new methylations at other sites. Despite being established by the same methyltransferase complex, the sperm methylome operates very differently from the oocyte methylome (new Ref 41). In oocytes, hypermethylated and hypomethylated domains are respectively correlated with active transcription units, and intergenic or inactive genomic regions(new Ref 42), while in the male germline methylation is established indiscriminately, at least apparently (new Ref 43,44). However, at present, the underlying transgenerational epigenetic mechanisms remain largely unknown.

Added in the text (lines 197-200 and 207-214, page 5)

 41. - Trends in Endocrinology & Metabolism, February 2020, Vol. 31, No. 2 https://doi.org/10.1016/j. tem.2019 10.0054  

Also, the mechanism by which dietary fonts of metabolites involved in epigenetic is unclear to me. Why are they protective, does this protection extend to all cancers, or a particular subset. Are these specific to trans-generational inheritance? That is what the title of the section implies.

But, the cited article refers to protective effects of nutrition on epigenetics beyond the gamete.

Nutrients and bioactive food components can either act directly by inhibiting epigenetic enzymes that catalyze DNA methylation or histone modifications, or by altering the availability of substrate necessary for those enzymatic reactions. This in turn modifies the expression of critical genes associated with physiologic and pathologic processes, including embryonic development, aging, and carcinogenesis.(new ref 42, 43)

42.Nelson VR, Nadeau JH. Transgenerational genetic effects. Epigenomics. 2010 Dec;2(6):797-806. 43.Bodden C, Hannan AJ, Reichelt AC. Diet-Induced Modification of the Sperm Epigenome Programs Metabolism and Behavior. Trends Endocrinol Metab. 2020 Feb;31(2):131-149.

There is a growing evidence that folate modulates anticarcinogenic properties through epigenetic changes, as low folate intake has been associated with DNA hypomethylation and increased risk of colorectal and pancreatic cancers. Diets rich in fruits and vegetables containing natural anti-oxidants may protect against cancer.

Certain dietary components (such as curcumin and retinoic acid) (45) may protect against cancer through miRNA regulation.

The potential epigenetic effects of several other nutritional components, mostly derived from vegetables (polyphenols, isothiocyanates, diallyl disulfide)  consist in the reduction in DNA hypermethylation of critical genes, finally resulting in tumor suppression. (46)

44.Thompson MD, Derse A, Ferey J, Reid M, Xie Y, Christ M, Chatterjee D, Nguyen C, Harasymowicz N, Guilak F, Moley KH, Davidson NO. Transgenerational impact of maternal obesogenic diet on offspring bile acid homeostasis and nonalcoholic fatty liver disease. Am J Physiol Endocrinol Metab. 2019 Apr 1;316(4):E674-E686.

45.Choi, S.W.; Friso, S. Epigenetics: A New Bridge between Nutrition and Health. Adv. Nutr. 2010, 1, 8–16.

  1. Davis, C.D.; Ross, S.A. Dietary components impact histone modifications and cancer risk. Nutr. Rev. 2007, 65, 88–94.
  2. Bishop, K.S.; Ferguson, L.R. The interaction between epigenetics, nutrition and the development of cancer. Nutrients 2015, 7, 922–947.

Dietary patterns, not only individual nutrients, also influence behavior and phenotype in offspring.

The traditional Mediterranean diet, rich in vegetables and fresh fruits, whole grains, seafood is associated with a reduced incidence of diet related cancers, for example the risk of breast cancer has been reported to be reduced in women who eat a Mediterranean diet supplemented with extra-virgin oil and mixed nuts. (47). Added in the text (lines 217-233, page 5)

I find that replication of the table from an already published review in the same family (MDPI) is inappropriate. The table should include new and additional data or references in order to add something to the field. Otherwise just cite the original reference.

The table has now been removed and the paper is cited as reference number 14.

Minor Points:

Spurious use of capital letters such as: nutrition transition in line 10, cancer and nutrition on line 20, globalization on line 41, syndemic on line 45, and food system on lines 52-53. “This review shortly analyses…” (line 14) should read This brief review analyses… Wrong selection of adjectives that render the phrase grammatically incorrect such as: “The number of humans living on the planet has incredibly increased ….” A better word selection would be “has significantly increased…” References are missing, such as the one in support of the sentence beginning on line 127 and finishing on line 129. Abbreviations such as those for the minerals indicated on line 177 should be explained, or, preceded by the full word, Calcium (Ca) for example.

The suggested changes have been made

Capital letter have now been removed as suggested

Abbreviations such as those for the minerals indicated on line 177 have been explained.

The reference in support of the sentence on line 127-129 has been added:

(new Ref. 22) - Sarah Whitmee, Andy Haines, Chris Beyrer, Frederick Boltz, Anthony G Capon, Braulio Ferreira de Souza Dias, Alex Ezeh, et al. The Rockefeller Foundation–Lancet Commission on planetary health Safeguarding human health in the Anthropocene epoch: report of The Rockefeller Foundation–Lancet Commission on planetary health. Published Online July 16, 2015 http://dx.doi.org/10.1016/ S0140-6736(15)60901-1

Reviewer 2 Report

I would like to thank the authors for the effort they made to write such good manuscript. This well written manuscript addresses an important topic of exploring different causes of carcinogenesis.

Author Response

The authors thank the reviewer for the valuable comments and suggestions.

Reviewer 3 Report

The authors shortly summarize the effect of climate change on crop production and so the biodiversity of food choices. Therefore, the dietary habit and nutrient intake will be dramatically affected.

In line 21, “…many times in the literature.”, please cite the literatures here; In line 25, “…some of the most diffuse types of cancer.”, it would be helpful to cite the relevant literatures here; In line 31-32, could you specify “human food habits dramatically changed” from what to what? In line 51-52, since this is a paper about nutrition, could you specify what is “macro- and micronutrient intake” respectively? In line 139, please specify what’s the meaning of “CHO diet”.

Author Response

The authors thank the reviewer for the valuable comments and suggestions.

Comments and Suggestions for Authors

The authors shortly summarize the effect of climate change on crop production and so the biodiversity of food choices. Therefore, the dietary habit and nutrient intake will be dramatically affected.

In line 21, “…many times in the literature.”, please cite the literatures here; In line 25, “…some of the most diffuse types of cancer.”, it would be helpful to cite the relevant literatures here; In line 31-32, could you specify “human food habits dramatically changed” from what to what? In line 51-52, since this is a paper about nutrition, could you specify what is “macro- and micronutrient intake” respectively? In line 139, please specify what’s the meaning of “CHO diet”.

Relevant literature has been added where required

References for the sentence on line 21:

  1. Mayne ST, Playdon MC, Rock CL. Diet, nutrition, and cancer: past, present and future. Nat Rev Clin Oncol. 2016 Aug;13(8):504-15.
  2. Shivappa N, Bosetti C, Zucchetto A, Serraino D, La Vecchia C, Hébert JR.Dietary inflammatory index and risk of pancreatic cancer in an Italian case-control study. Br J Nutr. 2015 Jan 28;113(2):292-8.

3. Sieri S, Pala V, Brighenti F, Agnoli C, Grioni S, Berrino F, Scazzina F, Palli D, Masala G, Vineis P, Sacerdote C, Tumino R, Giurdanella MC, Mattiello A, Panico S, Krogh V. High glycemic diet and breast cancer occurrence in the Italian EPIC cohort. Nutr Metab Cardiovasc Dis. 2013 Jul;23(7):628-34 

References for the sentence on line 25:

4 Andreescu N, Puiu M, Niculescu M. Effects of Dietary Nutrients on Epigenetic Changes in Cancer. Methods Mol Biol. 2018;1856:121-139. 5. Zhai T, Li S, Hu W, Li D, Leng S. Potential Micronutrients and Phytochemicals against the Pathogenesis of Chronic Obstructive Pulmonary Disease and Lung Cancer. Nutrients. 2018 Jun 25;10(7).  

Macronutrients are: carbohydrateprotein, and lipids; micronutrients are vitamins and minerals; this has now been specified in the text Added in the text (lines 53 - 54, page 2)

Line 31 – 32 “ …. from local fresh foods (vegetable, tuber, or animal-source foods) and home-cooked basic commodities to packaged and processed ready-to-eat or ready-to-heat food” ..... added in the text (new Ref 9,10)

9. Ronto R, Wu JH, Singh GM. The global nutrition transition: trends, disease burdens and policy interventions. Public Health Nutr. 2018 Aug;21(12):2267-2270.  10. Crittenden AN, Schnorr SL. Current views on hunter-gatherer nutrition and the evolution of the human diet. Am J Phys Anthropol. 2017 Jan;162 Suppl 63:84-109.